# ENTROPHY: USER INTERACTION DATA FROM LIVE ENTERPRISE WORKFLOWS FOR REALISTIC MODEL EVALUATION

## ABSTRACT

AI-driven automation for complex enterprise workflows faces significant hurdles due to the lack of publicly available datasets that realistically capture how business processes unfold – interaction by interaction – within actual production environments. Existing datasets are typically synthetic, confined to sandbox settings, or restricted to short web-based processes, limiting the preparedness of AI models for real-world complexities encountered in finance, legal, HR, and other critical domains. To bridge this gap, we introduce **ENTROPHY**, the first openly available dataset capturing detailed, end-to-end recordings of authentic enterprise processes. Experienced finance, legal, and HR professionals conducted 283 real-world workflow executions, totaling 33 hours of interactive activity across 19 diverse platforms spanning modern SaaS tools, web pages, and legacy desktop software. Each digital interaction is comprehensively logged alongside rich UI context and visual screen captures. Crucially, **ENTROPHY** captures not just structured process flows (and the overlap between them), but also the authentic, often messy dynamics of human work: multitasking, interruptions, off-process behaviors, and natural variability across users. By emphasizing fine-grained user interactions as a primary data modality, **ENTROPHY** provides a foundation for building AI systems capable of handling the nuances of real-world work in enterprise environments. As a first application, we benchmark frontier language models on workflow classification and boundary-accurate stream segmentation tasks, both central to enterprise automation, revealing substantial headroom for improvement. We make the dataset available at: https://kaggle.com/datasets/94647fd0bb51dff501a463674a2314627cdaf8c76d41b0 93c333b608459e017e.

## 1 INTRODUCTION

There is growing interest in deploying AI systems within enterprise environments to automate business processes (Grohs et al., 2024; Muthusamy et al., 2023; Rizk et al., 2024; Vidgof et al., 2023; Wornow et al., 2024a). These processes are not incidental; they underpin how organizations actually get work done, from onboarding an employee to closing the financial books. While it is tempting to model these workflows at a high level, doing so overlooks the complexity and variability that define them in practice. On paper, business processes appear well-defined: structured into clear sequences of steps, embedded with business semantics (e.g., contract terms, supplier data), executed across multiple applications, and tied to concrete business outcomes. Yet, actual execution is rarely clean. Teams skip or reorder steps, use tools inconsistently, or pursue alternate execution paths depending on local context. Variability also stems from differences in geography or team conventions. AI systems must therefore be trained to operate at the level where enterprise work actually unfolds: in the moment-by-moment digital interactions between people and computers. This includes not only the structured logic of workflows, but also the execution-level dynamics that arise in practice, such as multitasking, context-switching, and unplanned detours. These dynamics introduce *variance* and *noise* in business processes, which are not anomalies but defining features of real-world enterprise work. Without explicitly modeling these effects, AI systems will fail to operate reliably in live environments (Kayali et al., 2025; Marreed et al., 2025; Vidgof et al., 2023; Xu et al., 2024).

Capturing this complexity requires logging the full stream of digital user interactions – mouse clicks and keystrokes – alongside on-screen metadata, which together reflect how real users engage with software systems to carry out processes. Most existing datasets fall short in three important ways. First, many are built synthetically from scripted scenarios or sandboxed environments and lack the behavioral realism of live production settings. Second, even when real users are involved, they often center on short consumer-facing processes or isolated web interactions, not enterprise-grade workflows. Third, they overwhelmingly emphasize browser-based applications, ignoring the legacy software that remains central to critical enterprise functions.

To address these limitations, we introduce **ENTROPHY**, a dataset of real-world business processes captured in production settings. It is sourced from trained finance, legal, and HR professionals performing their actual day-to-day work in an organization. **ENTROPHY** comprises 33 hours of user activity, spanning 283 annotated instances across 24 unique processes and 19 platforms, including both modern (web) applications and legacy desktop tools. Each event in the dataset represents a digital user interaction (e.g., click, data entry, or hotkey), enriched with contextual metadata such as screen title, field label, application identity, timestamp, and a screenshot. **ENTROPHY** is unique among existing datasets in several key ways, as it:

1. Captures complete business processes tied to real outcomes, not simulated workflows or isolated micro-processes.

2. Spans both legacy desktop tools and modern browser applications, enabling modeling across the full enterprise software stack.

3. Contains detailed, semantically grounded logs of digital user interactions, recorded in live production settings as professionals perform their actual day-to-day work.

4. Contains both the structural backbone of enterprise workflows and the behavioral variability seen in practice, such as multitasking, interruptions, and alternate execution paths.

5. Covers processes of substantial duration, all of which last several minutes, with some extending well beyond 20 minutes of continuous activity (over $40\times$ longer than comparable datasets).

These features make **ENTROPHY** a valuable resource for advancing enterprise AI, where models have to be evaluated in real environments that involve long, multimodal workflows with complex and variable sequences. Such settings demand capabilities such as long-context reasoning, decision-making under uncertainty, and navigating diverse UIs – core challenges currently pursued in ML research through long-context LLMs, instruction-following agents, and tool-using systems. By capturing the full complexity of enterprise workflows, **ENTROPHY** offers a high-fidelity testbed for developing robust models and establishing benchmarks of their performance in realistic work settings.

As a first application, we provide benchmark evaluations using state-of-the-art language models (Claude 3.5 Haiku (Anthropic, 2024), DeepSeek-R1 (DeepSeek-AI, 2025), Gemini 2.5 Flash (Kavukcuoglu, 2024), GPT-4.1 (OpenAI), Qwen3-32B (Yang et al., 2025)) on workflow classification and segmentation tasks derived from the dataset. Overall, we find that current frontier language models achieve at most ~70% accuracy on the evaluated tasks, with errors concentrated on structurally similar but semantically distinct workflows. While promising, this level of performance remains insufficient for reliable deployment in enterprise settings, where higher accuracy and consistency are critical for downstream utility and trust, underscoring the importance of datasets like **ENTROPHY** in making these limitations visible.

The remainder of the paper is organized as follows: Sec. 2 compares **ENTROPHY** to existing datasets. Sec. 3 discusses the challenges of collecting real-world digital interaction data and our approach to overcoming them. Sec. 4 provides a detailed overview of our dataset. Sec. 5 demonstrates downstream tasks enabled by the dataset. We conclude in Sec. 6. Additional technical details are provided in the supplementary materials.

## 2 RELATED WORK

Many datasets have been curated to evaluate models on their ability to understand and automate digital processes. We briefly survey the relevant literature, organizing it into two categories based on process provenance: synthetically generated processes and processes sourced from real-world

Table 1: Comparison of **ENTROPHY** with existing workflow datasets.

| Benchmark | Process Sourcing | # of Instances (# of Processes) | # of Apps | Avg Steps/Instance |
|---|---|---|---|---|
| WONDERBREAD (Wornow et al., 2024b) | Synthetic | 2,928 (598) | 4 | 7.8 |
| WebArena (Zhou et al., 2023) | Synthetic | 812 (241) | 4 | – |
| VisualWebArena (Koh et al., 2024) | Synthetic | 910 (314) | 3 | – |
| OmniAct (Kapoor et al., 2024) | Synthetic | 9,802 (–) | 65 | – |
| REAL (Garg et al., 2025) | Synthetic | 112 (–) | 11 | – |
| WorkArena (Drouin et al., 2024) | Real | 19,912 (33) | 1 | – |
| OSWorld (Xie et al., 2024) | Real | 369 (–) | 10 | – |
| Mind2Web (Deng et al., 2023) | Real | 2,350 (–) | 31 | 7.3 |
| **ENTROPHY** (this work) | Real | 283 (24) | 19 | 178 |

settings. Then, we provide background on process mining and interaction intelligence work related to the broader understanding and improving of digital workflows.

**Synthetic processes:** Artificially generating processes in sandbox environments enables large-scale dataset collection and controlled model evaluation. While effective for scale, the collected processes tend to be significantly shorter and less complex than real-world enterprise processes. For example, WebArena (Zhou et al., 2023), REAL (Garg et al., 2025), and VisualWebArena (Koh et al., 2024) provide dynamic environments for agents to interact with websites to complete a curated set of processes. Despite this flexibility, the websites are simplified clones of their real-world counterparts, and most processes can be completed in a dozen steps or fewer. Other datasets, such as OmniAct (Kapoor et al., 2024), curate larger collections based on real-world applications, but define processes retrospectively through post hoc human annotation. Finally, benchmarks focused on process mining like WONDERBREAD (Wornow et al., 2024b) also rely on WebArena's set of synthetic processes and sandboxed setup.

**Real-world processes:** Curating data from real-world use cases is inherently more challenging due to cost and privacy constraints. Nonetheless, benchmarks grounded in such data offer a more accurate reflection of model performance in enterprise settings. Compared to synthetic datasets, real-world processes tend to involve longer, more complex processes that span multiple applications. However, these datasets are often significantly smaller, lack dynamic environments for evaluation, and provide shallower annotations than their synthetic counterparts. For example, Mind2Web (Deng et al., 2023) contains 2,350 processes sourced from real-world websites, but does not come with a dynamic execution environment for evaluating models. WorkArena (Drouin et al., 2024) and OSWorld (Xie et al., 2024) both offer dynamic execution environments with real-world processes, but the former is limited to one application (ServiceNow) while the latter only offers one demonstration per process.

In contrast to the above datasets, ours uniquely combines fine-grained, application-agnostic interactions with process-level annotations of real-world workflows collected at scale, enabling versatile use cases and more generalizable insights into user behavior. We compare the different datasets in Tab. 1. Notably, an average workflow in **ENTROPHY** contains roughly $10\times$ more interaction steps than those found in comparable datasets.

**Process mining and interaction intelligence:** Industry has long explored the problem of understanding and improving digital workflows through tools such as process mining and interaction analytics. Process mining focuses on extracting structured process models from event logs generated by enterprise systems (Reinkemeyer, 2020), enabling organizations to visualize bottlenecks and inefficiencies in processes. These traditional approaches rely heavily on system logs and structured backend data, but miss the fine-grained realities of human behavior during process execution. At the other end, "digital interaction intelligence" tools (Modi & Kumar, 2024) record clicks and keystrokes for automation discovery and performance tuning (Bru & Claes, 2018; Modi & Kumar, 2024), yet their data are proprietary. **ENTROPHY** bridges this gap by offering open access to high-fidelity interaction data collected in live enterprise settings, supporting research on modeling and understanding real-world business processes.

## 3 COLLECTING DIGITAL INTERACTION DATA

Modern enterprise workflows unfold through thousands of fine-grained human-computer interactions that span legacy bespoke systems, desktop tools, and modern web applications. Capturing these

interactions at scale, without disrupting end-user experience or violating privacy regulations, requires a collection stack that is *application-agnostic*, *light-weight*, and *privacy-preserving*. To meet these requirements, we built a proprietary data collection system designed to operate reliably across varied enterprise environments while maintaining both privacy and performance. In this section, we define what constitutes a digital interaction, describe its representation in the dataset, and outline the core challenges in enterprise-scale data capture.

## 3.1 ANATOMY OF A DIGITAL INTERACTION

A digital interaction is an atomic unit of observed user activity. Each time a person interacts with software – whether within an application, a browser, or the operating system – the event is recorded as one of three types: *Clicks* (events where a user clicks on a button, checkbox, icon, menu option, and other such clickable GUI elements), *Data entries* (events where a user continuously enters data, typically with keystrokes, into a textbox or any other similar data entry field), and *Hotkeys* (events which may look like data entry, as they are keystrokes, but actually have a richer semantic meaning, such as Ctrl+C).

All actions performed by a user are mapped into these three fundamental types of interactions. We then derive complex interactions that are a combination of these types. For example, *navigation* is when a user performs a sequence of basic actions that result in movement within or across applications. The full structure of a digital interaction as captured by our tool and represented in **ENTROPHY** is shown in Tab. 2. It includes a high-level category (class name, e.g., *navigation*), a fine-grained subtype (subclass name), and a semantic description, along with contextual metadata such as timestamp, application name, and screen name. Each process workflow in the dataset is a series of hundreds of such digital interactions, which we present in Sec. 4.

Table 2: Fields comprising a digital interaction, with representative values drawn from **ENTROPHY**.

| Field Name | Example Value | Description |
|---|---|---|
| Process Instance UUID | 67262701-6d66-492d-af78-51789e08572e | Unique identifier for the process instance this interaction comes from. |
| Process Name | Invoice creation | Manually configured name for the business process being carried out. |
| Interaction UUID | 9c2c8808-9a31-4e05-b151-e6d17ec6ae4d | Unique identifier for the individual digital interaction. This is unique across the dataset. |
| Class Name | Application Field Input | High-level category assigned to the interaction, based on a deterministic rule set. |
| Subclass Name | Edit Field | Subcategory of the class name, also determined by a rule-based system. |
| Description | Editing an application field - invoice | Description of the interaction. |
| Timestamp | 2025-04-18 17:01:16.466 | When the user performed the interaction, in UTC. |
| Application Name | 4814618-sb1.app.netsuite.com | Name of the application where the interaction occurred. For web applications, this is a URL. |
| Screenshot Name | Screen-2025-04-18T17-01-16.466E480.png | Name of a PNG screenshot associated with the interaction, or Null if none exists. |
| Screen Name | Invoice Net Suite <name> Private Limited | Descriptive screen name derived from the window title or on-screen headers. Subject to PII filtering. |
| Interaction Type | Click | Action that a user took on a GUI element, i.e., Click or Typing. |
| Interaction Value | LB Down | Input signal corresponding to the action. |
| Interaction Coordinates | {'x': 159, 'y': 484} | Screen coordinates (in pixels) where the interaction occurred. |
| Field Name | INVOICE* | Name associated with the interacted field. |
| Field Value | 56202596 | Value associated with the interaction. |
| Time Spent | 0.643 | Time spent interacting with the element, in seconds. |

## 3.2 CHALLENGES IN DIGITAL INTERACTION COLLECTION FOR THE ENTERPRISE

Capturing digital interaction data in live enterprise settings, at scale, and in compliance with privacy requirements, presents significant technical and operational challenges listed below.

**Application diversity:** Enterprise environments contain a heterogeneous mix of software, including legacy systems, mainframe applications, and modern web platforms. Many are decades old or mission-critical, where making modifications for data capture is infeasible. Tools like the Windows

Accessibility API can be used to try and observe digital interactions, but such APIs often fail to generalize and can degrade both application performance and data quality. Even modern web applications may exhibit lag when instrumented. A robust collection system must extract high-fidelity interaction data across diverse applications and web platforms without impacting user experience.

**Regulatory compliance:** Regulations such as GDPR and HIPAA require safeguards against the collection and retention of personally identifiable information (PII). For instance, under GDPR, a protected customer may request deletion of their personal data. Consequently, data capture systems must be designed with semantic awareness to identify and manage sensitive content proactively, incorporating mechanisms for real-time redaction, secure storage, and compliant data deletion.

**User privacy:** Beyond formal compliance, employees reasonably expect that sensitive personal information, such as names, emails, or other internal identifiers will be masked or excluded from any captured data. This expectation stems not only from privacy norms but also from a desire to avoid unwanted exposure, profiling, or workplace surveillance. While omitting entire screens is technically straightforward, suppressing individual data entry fields requires real-time semantic parsing to distinguish between innocuous and sensitive inputs. In dynamic, multi-application environments, this requires lightweight on-device logic that can identify and redact sensitive information before it is stored.

**Performance and timing constraints:** Capturing digital interactions at high fidelity involves operations such as API polling, screen parsing, and metadata extraction – tasks that are resource intensive but must not interfere with the responsiveness of the user's machine. In production settings, even minor slowdowns are often unacceptable. Compounding this, user interactions often unfold rapidly, with inputs immediately followed by interface transitions. If the capture system lags even briefly, contextual information can vanish before it is recorded. To ensure completeness of the interaction trace, capture must occur within milliseconds of the interaction, while staying within strict CPU and memory constraints.

### 3.3 Our Approach: Lightweight Vision Models

To overcome the challenges outlined above, we developed a real-time data collection system powered by custom, lightweight vision models. The dataset presented in this paper was collected and labeled using these models, which run directly on end-user machines to enable accurate, low-latency capture of digital interactions, without requiring access to application source code or internal APIs.

Our system performs three core tasks in real time:

- Detect visible UI elements across arbitrary applications and web platforms.
- Link semantically related elements through key-value pairs.
- Record user interactions as structured digital events, grounded in UI context and human-readable semantics.

The vision backbone is a YOLO-style object detector trained on an internally annotated dataset of enterprise UI components. To ensure deployment on typical enterprise machines, we applied model compression techniques such as structured pruning and knowledge distillation from a larger teacher model, reducing size and latency while preserving detection accuracy. At runtime, the model anchors user actions to detected components and on-screen text, which are mapped into the three fundamental interaction types described in Sec. 3.1 and aggregated into structured digital events.

Unlike large-scale vision–language models (e.g., ViT-based systems with hundreds of millions of parameters), our models are compact enough to run efficiently on low-spec enterprise machines, yet robust enough to generalize across legacy applications, web interfaces, and mainframe terminals. This visual-first, platform-agnostic design avoids brittle integrations, supports systems that cannot be instrumented through conventional means, and ensures all processing remains local, with sensitive fields masked at capture time. Visual data is retained only when users explicitly contribute labeled workflow samples, since these are easiest for business users to validate.

By combining model efficiency, semantic fidelity, and real-world robustness, our system enables scalable, privacy-aware digital interaction capture in live enterprise environments – a capability we believe to be novel.

# 4 THE DATASET

To build a dataset that reflects how enterprise work is actually performed, we recruited 9 trained volunteers from within our organization with their explicit consent – 4 in finance, 1 in legal, and 4 in HR. These volunteers repeatedly carried out real-world processes over 5 working days which contributed to this dataset. Each participant had at least four years of experience in their domain, and the processes they executed mirror the kinds of day-to-day processes routinely performed by professionals in large enterprises. Drawing on our team's significant expertise in this space, we ensured that the workflows captured in **ENTROPHY** were not abstract simulations, but faithful representations of business processes as they occur in practice, including the variability, interruptions, and natural execution patterns typical of production environments.

Over the course of the study, we recorded 108 instances of 11 unique processes in finance, 102 instances of 7 processes in legal, and 73 instances of 6 processes in HR, totaling 24 distinct process types (see Tab. 3 for an overview). These workflows span approximately 33 hours of activity and involve 19 distinct applications and web domains, including Outlook, Excel, Word, NetSuite, DocuSign, Greenhouse, and Zoho. All processes were executed end-to-end using the same applications, screens, and workflows found in real enterprise environments, with only minor modifications to protect personally identifiable information:

**Staging environments:** To safeguard sensitive information, all processes were executed on 'staging' instances of the *same* enterprise applications used by the teams in their actual work.

**Synthetic entities:** To protect privacy, all references to companies, contracts, employees, job applicants, and job openings were replaced with artificial entities that bear no resemblance to real individuals or organizations. Importantly, no customer data was used at any stage of the collection process.

**Application filtering:** Applications that could not be replicated in a staging environment were excluded from the dataset. These included platforms such as government portals for credential verification, bank websites for account validation, and external job boards used for sourcing applicants.

Each participant's workstation was instrumented with our data collection tool. Participants were informed when recording was active. At the start of a workflow, they selected its process name from a dropdown menu and initiated recording by pressing Start; upon completion, they pressed Stop. During the session, the tool logged all interactions within a predefined set of whitelisted applications and tagged each event with the selected process name. Screenshots were captured only on clicks or non-printable keystrokes (e.g., Enter, Tab, function keys), and were throttled to a maximum of two frames per second. Further details on the process recording procedure are provided in App. A.

## 4.1 COMPLEXITY IN ENTERPRISE WORKFLOWS

Many processes in **ENTROPHY** span hundreds of digital interaction steps across multiple applications and persist for several minutes of continuous activity, sometimes even exceeding 20 minutes per instance. These processes reflect the full operational complexity of real enterprise work, manifesting as overlap, variation, and noise, which we detail below.

**Overlap:** Distinct, long processes, even within the same team, may have significant overlap in their sequences of interactions, making their differentiation harder (see App. B.4).

Process duration alone is notable, but what sets the dataset apart from existing ones is the variability embedded in each process. As shown in Tab. 3, the standard deviation in process durations is often large relative to the mean, revealing underlying variability as a key source of execution complexity.

**Variation:** We define variations as differences in workflow steps or sequences that are intrinsic to how real-world processes are performed. These may result from process design, team practices, domain-specific exceptions, or individual user strategies. In our dataset, variations occur in two primary ways:

- **Across processes:** Different processes appear with varying frequencies and durations depending on business context and operational needs. For example, a Master Service Agreement (MSA) is typically created once for a client and reused, whereas a Statement of Work (SOW) is generated

Table 3: Summary of the **ENTROPHY** dataset.

| | Process Name | # of Instances | # of Apps | Avg Steps/Instance | Time (mean ± std) [s] |
|---|---|---|---|---|---|
| **Finance** | Invoice processing | 3 | 6 | 307 | $756 \pm 587$ |
| | Purchase order creation | 10 | 4 | 140 | $368 \pm 54$ |
| | Vendor onboarding | 3 | 6 | 408 | $949 \pm 248$ |
| | Contract updates and forwarding | 21 | 7 | 253 | $718 \pm 212$ |
| | Financial review of contract | 15 | 8 | 39 | $162 \pm 57$ |
| | Invoice creation | 13 | 7 | 231 | $547 \pm 189$ |
| | Invoice dispatch | 14 | 10 | 113 | $296 \pm 107$ |
| | Payment collection | 14 | 6 | 67 | $158 \pm 75$ |
| | Revenue accounting | 3 | 6 | 549 | $921 \pm 615$ |
| | Invoice accounting | 9 | 3 | 150 | $449 \pm 144$ |
| | Invoice payment | 3 | 5 | 308 | $666 \pm 387$ |
| **Legal** | Contract kickoff | 22 | 7 | 129 | $318 \pm 237$ |
| | First pass contract review | 1 | 4 | 128 | 402 |
| | MSA contract iteration | 1 | 3 | 261 | 660 |
| | Legal finalization and MSA execution | 1 | 4 | 299 | 859 |
| | SOW kickoff and order form creation | 20 | 4 | 179 | $371 \pm 170$ |
| | SOW iteration | 20 | 5 | 148 | $406 \pm 184$ |
| | Legal finalization and SOW execution | 37 | 8 | 97 | $277 \pm 166$ |
| **HR** | Candidate sourcing and screening | 14 | 9 | 115 | $320 \pm 149$ |
| | Job creation | 21 | 9 | 259 | $520 \pm 158$ |
| | Employment letter generation | 15 | 7 | 199 | $338 \pm 75$ |
| | Employment record reporting | 4 | 4 | 730 | $1014 \pm 305$ |
| | Monthly leave reporting | 4 | 4 | 709 | $1167 \pm 154$ |
| | Employee offboarding | 15 | 3 | 132 | $268 \pm 89$ |

for each new project. This leads to fewer recorded instances of MSA-related processes (e.g., MSA Contract Iteration) and many more of SOW-related ones (e.g., SOW Iterations). Similarly, accounts payable processes are often executed in batches, resulting in fewer but longer instances, while accounts receivable processes are typically handled one at a time, producing shorter but more frequent instances.

- **Within processes:** Even for the same nominal workflow, execution paths can differ due to conditional business logic, user preferences, or local optimizations. A contract review may include additional approval loops in some instances but follow a direct path in others. In talent acquisition, a recruiter might create a job post from scratch or clone an existing one. Importantly, there is no canonical definition of a process and hence the same process could be executed in many different ways across teams and organizations.

These variations are reflected not only in the number of interactions and applications used, but also in the time required to complete each instance.

**Noise:** We define noise as user actions captured during process execution that are unrelated to the process objective. Although not part of the intended workflow, such actions naturally occur in live enterprise settings and reflect the actual conditions under which work is done. Noise manifests in several forms: distractions, such as briefly visiting unrelated websites (e.g., news or sports); personal errands, like checking travel sites or reviewing non-work documents; email scanning, where users browse irrelevant messages while looking for process-relevant content; context switches caused by external interruptions (e.g., chat pings, meeting reminders); and breaks or idle time, during which a process is paused without further interaction.

The standard deviations reported in Tab. 3 may stem from these noise patterns rather than workflow structure alone. While extraneous to the process, these behaviors are not anomalous – they are part of the reality of enterprise work. For modeling, the ability to distinguish between relevant and incidental actions is essential for building robust AI systems that function effectively in real-world conditions. We provide a deeper dive into these dynamics, including a visual comparison of multiple executions of the same process, in App. A.4.

## 5 BENCHMARK TASKS

To demonstrate the utility of **ENTROPHY** and to establish initial performance baselines, we introduce the following two benchmark tasks. Code is available as a zip archive in the supplementary materials.

**Workflow classification:** Given a sequence of interactions corresponding to a single complete workflow instance, the task is to classify the sequence into one of predefined workflow classes, e.g., "Invoice processing". Since there is no canonical definition of any process across teams or organizations, being able to map a sequence of user activity to a process is an essential requirement on this data. This is key to deal with overlap, variance, and noise in how teams execute processes.

**Workflow segmentation:** This task requires models to identify the boundaries between distinct, consecutive workflow instances that have been concatenated into a single, long sequence of user interactions. Accurately locating the *start* and *end* of each workflow instance is indispensable in practice: business metrics such as average-handling-time, turnaround-time, among others are computed over these very intervals, so boundary errors propagate directly to managerial dashboards and, ultimately, to operational decisions.

Together, these tasks also supply building blocks for downstream agents: classification enables intent recognition and retrieval of relevant demonstrations, while segmentation equips agents to reset internal state, trigger new plans, and manage context switches in continuous activity streams.

For both tasks, we test the ability of foundation models to understand and reason about the structured sequences of user interactions present in our dataset. We evaluated several frontier LLMs, including non-reasoning models such as GPT-4.1 and Claude-3.5-Haiku, as well as reasoning models such as Qwen3-32B, Gemini-2.5-Flash, and DeepSeek-R1. Although our main benchmarks rely on structured interaction sequences for comparability across models, `ENTROPHY` is inherently multimodal, and we therefore also explored the use of screenshots alongside interactions. For additional details and extended experimental results, we refer the reader to App. B.

## 5.1 WORKFLOW CLASSIFICATION

This task tests the model's ability to recognize high-level patterns and semantic cues within the interaction data that are characteristic of specific business processes. We evaluated the models in a *zero-shot setting*, i.e., the model is asked to classify a workflow instance without any prior information about the workflows. The prompt includes only the interaction sequence and the list of possible workflow classes. For each model, we process the entire dataset, feeding one workflow instance at a time. The model output – the predicted workflow class – is then compared against the true label.

In Fig. 1, we present the classification accuracy of the frontier models across different domains. Notably, while models reach their highest accuracy on HR workflows (in the 70% range), this level remains insufficient for enterprise use, and performance drops even further in finance and legal. To understand the substantial drop in performance in the legal domain, we explore whether it is due to the models' limited domain-specific knowledge or the intrinsic similarity of legal workflows, which may make them harder to differentiate. To assess this, we represent each workflow instance as a dense vector using OpenAI's `text-embedding-3-large` model (OpenAI, 2024) and compute pairwise cosine similarities. As shown in App. B.4, based on the observed patterns in embedding similarity, we conclude that the reduced classification performance in the legal domain is likely attributable to the high intrinsic similarity among its workflows, rather than solely to a lack of domain-specific knowledge in the models. This overlap in semantic structure across different legal processes makes them more challenging to distinguish, thereby limiting the models' ability to effectively classify them. In contrast, HR workflows exhibit clearer process-level distinctions, contributing to higher model performance in that domain.

## 5.2 WORKFLOW SEGMENTATION

Given a concatenated sequence of interactions, the model must output the start and end indices for each constituent workflow. This task evaluates a model's ability to detect shifts in context and activity patterns that signal the conclusion of one process and the beginning of another, a crucial capability for understanding multi-stage business operations or analyzing long, uninterrupted user sessions.

For each domain (HR, legal, and finance), we construct a total of 100 input samples by concatenating several (e.g., 2 to 5) randomly selected workflow instances. The models are provided with this concatenated sequence of interactions and a list of all possible workflow process *definitions*. The prompt instructs the model to output a JSON array specifying the start/end indices for each identified workflow segment. The evaluation compares these predicted boundaries against the true boundaries.

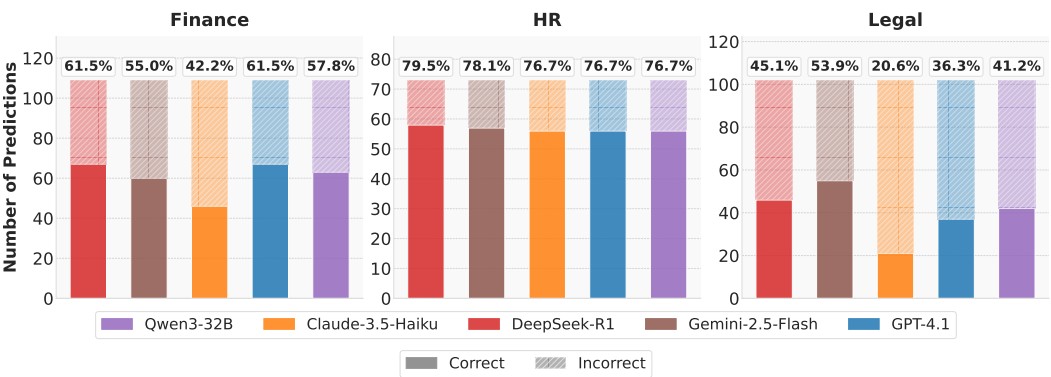

Figure 1: Zero-shot performance of frontier models on workflow classification. The bar height corresponds to the total number of workflow instances, while the bar fill(%) indicates correctness.

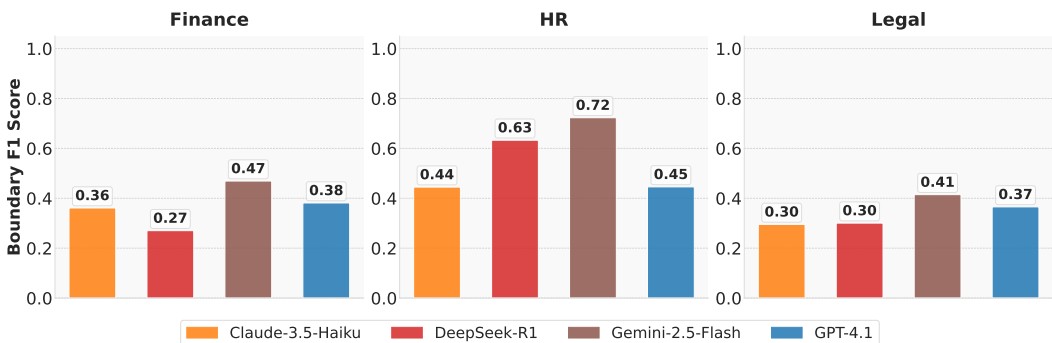

Figure 2: Zero-shot performance of frontier models on our workflow segmentation benchmark.

We summarize our results in Fig. 2. From our evaluations, we note two consistent patterns. **(i) Limited accuracy:** The best model, Gemini-2.5-Flash, reaches an F1 score of $0.72$ on HR, but only $0.47$ and $0.41$ on finance and legal, respectively. All other models fall below these numbers, confirming that precise process boundary detection remains a challenge in a zero-shot setting. **(ii) Recall $\gg$ precision:** From App. B.2, we see that every model shows substantially higher recall than precision, indicating a tendency to over-segment (i.e., insert spurious cuts). While generous recall mitigates missed workflows, low precision inflates duration estimates and therefore distorts business metrics.

Fine-tuning could narrow this gap, yet doing so requires exactly the sort of interaction-level data that is scarce in the public domain; **ENTROPHY** fills that void. Because process definitions vary across teams and industries, we anticipate that effective solutions will need either domain-adaptive training or novel architectures tailored to long, noisy action streams.

## 6 CONCLUDING REMARKS

**ENTROPHY** is the first public dataset to capture full-length enterprise workflows at interaction-level fidelity. Spanning 283 workflows, 33 hours of finance, legal, and HR activity across 19 applications and web domains, it preserves the multitasking, interruptions, and path variability absent from synthetic sets. Baseline experiments show that frontier language models have substantial headroom to improve. We release the data to spur work on workflow-aware AI models and agents. In contrast to traditional methods such as qualitative interviews or system logs, this dataset opens up a computational lens for understanding how work gets done. Just as web interaction logs unlocked breakthroughs in search and recommendation, we believe digital user interaction data will be foundational for building AI-native systems that understand and automate enterprise work.

**Ethics statement:** Our study involved nine internal subject matter experts performing representative enterprise workflows. Participation was voluntary, unpaid, and based on documented informed consent. As our organization does not have a formal IRB, we followed an internal review process: peer observers ensured transparency during recruitment, informed consent was documented in writing, and an independent internal audit reviewed all data for privacy risks. All personally identifiable information was removed prior to release, and it is not possible to identify any real individuals, organizations, or records from the data.

**Reproducibility statement:** Sec. 5 outlines the experimental setup and results, and App. B of the supplementary materials details the compute environment, prompts, and evaluation procedures. The dataset is available on Kaggle, and we include a zip archive in the supplementary materials with all code and scripts needed to reproduce our results.

**LLM usage:** Large language models were used as baseline systems in our experiments and as general-purpose writing assistants (e.g., grammar polishing, finding synonyms). They were not involved in research ideation or in generating dataset content. The authors take full responsibility for all text and results.

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

## SUPPLEMENTARY MATERIAL
## Entrophy: User interaction data from live enterprise workflows for realistic model evaluation

## A  DATASET

### A.1  DETAILS OF RELEASED DATASET

The released dataset is organized by domain – Finance, HR, and Legal – with each domain comprising a JSON file of digital interactions, structured as described in Tab. 2 of the main text, alongside a corresponding folder of screenshots. In total, the dataset contains approximately 50,300 interaction records and 37,500 screenshots. To safeguard privacy, all screenshots have been processed to blur employee faces, email addresses, full names, and other sensitive content. A comparable redaction process has been applied to the interaction logs. The dataset is made available under the CC BY-NC-SA 4.0 license (Commons, 2013).

### A.2  PROCESS RECORDING WORKFLOW

Data collection proceeded in five structured phases: (1) selecting volunteers from the Finance, HR, and Legal teams, (2) briefing them on the data collection goals and methodology, (3) identifying and defining candidate processes for recording, (4) executing those processes in sandboxed environments using our custom agent, and (5) obtaining informed consent for data use and publication. Each team completed these steps over approximately five days. We provide a brief description of each step below.

**Volunteer selection:** Subject matter experts (SMEs) were chosen based on their domain expertise, familiarity with relevant enterprise software, and willingness to participate. Within each team, process experts first defined representative workflows, which the selected SMEs then executed during the data collection phase.

**SME Instructions:** SME machines were pre-configured by IT with our lightweight data collection agent, designed to operate only within sandboxed environments to prevent capture of any personal or sensitive enterprise data. SMEs received both written documentation and in-person training on how to use the tool, including how to start, stop, and review recordings. Each participant completed a set of dry runs to ensure technical readiness and familiarity with the workflow.

**Process Definitions:** Process experts within each team curated a list of representative workflows, which were then documented and executed by the selected SMEs. These workflows were reviewed by the research team for suitability. Tasks that relied heavily on verbal communication, such as phone calls or meetings, were excluded. However, SMEs were encouraged to include natural variations and edge cases to reflect the full spectrum of real-world execution.

**Recording:** The data collection agent operated similarly to screen recording software, allowing users to initiate, pause, and stop sessions at will. SMEs were instructed to work as naturally as possible during recordings, including accommodating spontaneous interruptions such as unrelated messages, personal errands, or breaks. This design choice intentionally preserved the noise and variability inherent in real-world digital work.

**Informed Consent:** After all recordings were completed, the research team sent formal email requests to participating SMEs for their informed consent to include their interaction data in **ENTROPHY**. All participants responded with affirmative consent via email, authorizing the inclusion of the captured data in the final public release.

### A.3  DESCRIPTION OF RECORDED PROCESSES

The table below provides expanded descriptions for each process included in our study.

Table 4: Names and descriptions for Finance, Legal, and HR processes.

| Modified Name | Description |
| --- | --- |
| Invoice processing | Review incoming invoices, verify them against purchase orders (POs), and route to stakeholders for approval. |
| Purchase order creation | Create a PO after obtaining approval for a purchase requisition. |
| Vendor onboarding | Register a new vendor and update their details in the accounting system. |
| Contract updates and forwarding | Update the Customer Relationship Management (CRM) system with contract and PO details, then forward the record to Finance for review. |
| Financial review of contract | Validate contract terms and PO details in the CRM, and approve or reject as needed. |
| Invoice creation | Generate a customer invoice in the accounting system using the approved PO. |
| Invoice dispatch | Send the finalized invoice to the customer, typically via email or a secure portal. |
| Payment collection | Record incoming payments and update the aging report to track outstanding balances. |
| Revenue accounting | Update internal revenue records with payment details and newly signed customer contracts. |
| Invoice accounting | Extract invoice data, compute taxes and adjustments, and prepare for payment processing. |
| Invoice payment | Schedule and complete payment, notify the vendor, and update payment records in the accounting system. |
| Contract kickoff | Review contract requests, check for an existing Master Service Agreement (MSA), and initiate or update the MSA accordingly. |
| First pass contract review | Conduct an initial review of the MSA, identifying clauses that require internal clarification or external discussion. |
| MSA contract iteration | Revise the MSA based on feedback from internal stakeholders and the customer's legal team. |
| Legal finalization and MSA execution | Finalize and digitally execute the MSA with all relevant parties. |
| SOW kickoff and order form creation | Create the order form and initiate the Statement of Work (SOW), ensuring correct linkage to the MSA and all key terms. |
| SOW iteration | Collaborate with the customer to refine and finalize the terms outlined in the SOW. |
| Legal finalization and SOW execution | Digitally execute the finalized SOW and forward it to Sales Operations for CRM updates. |
| Candidate sourcing and screening | Upload resumes to the recruiting portal, assess candidate fit, and progress suitable profiles to the interview stage. |
| Job creation | Create or duplicate a job in the recruiting portal, ensuring that the job description is accurate and complete. |
| Employment letter generation | Generate a standardized employment letter, apply necessary updates, and digitally sign the document. |
| Employment record reporting | Extract employment data, reconcile with employee exit records, and generate a consolidated report. |
| Monthly leave reporting | Download employee leave records from the human resources portal, validate the entries, adjust discrepancies, and publish for compliance checks. |
| Employee offboarding | Process an employee exit by managing access rights and generating formal resignation and relieving letters. |

## A.4 EXAMPLE PROCESS: FINANCIAL REVIEW OF CONTRACT

Sales contracts are commonly generated when a business sells a product or service to another business or consumer. In business-to-business (B2B) contexts, these contracts typically require formal review before execution. The example described here, 'Financial review of contract', illustrates a common process in B2B software-as-a-service (SaaS) transactions. In this context, the buyer issues one or more purchase orders, accompanied by supporting documents, and the seller responds with an approved contract fulfilling those terms. The contract and accompanying materials are reviewed to ensure compliance with business policies and statutory requirements, and to detect errors such as incorrect pricing. For example, a review may verify that no product exceeds a 30% discount cap or that the correct sales taxes are applied. We examine three instances of such a process, see also Fig. 3.

**Variant 1:** This instance begins with the user receiving an email containing a link to a contract form. Clicking the link opens the Customer Relationship Management (CRM) system (Salesforce). The user reviews the supporting documents, two purchase orders followed by deal-split information, then creates a new purchase order and enters the required details. The process concludes with the approval of the order within the CRM.

**Variant 2:** This instance also begins with an email linking to the CRM system. The user first reviews deal-split information, then examines the purchase orders, followed by an additional review of the billing schedule. An error occurs when the user attempts to approve the order without first creating a new purchase order, triggering a system message. The user is prompted to enter the required order

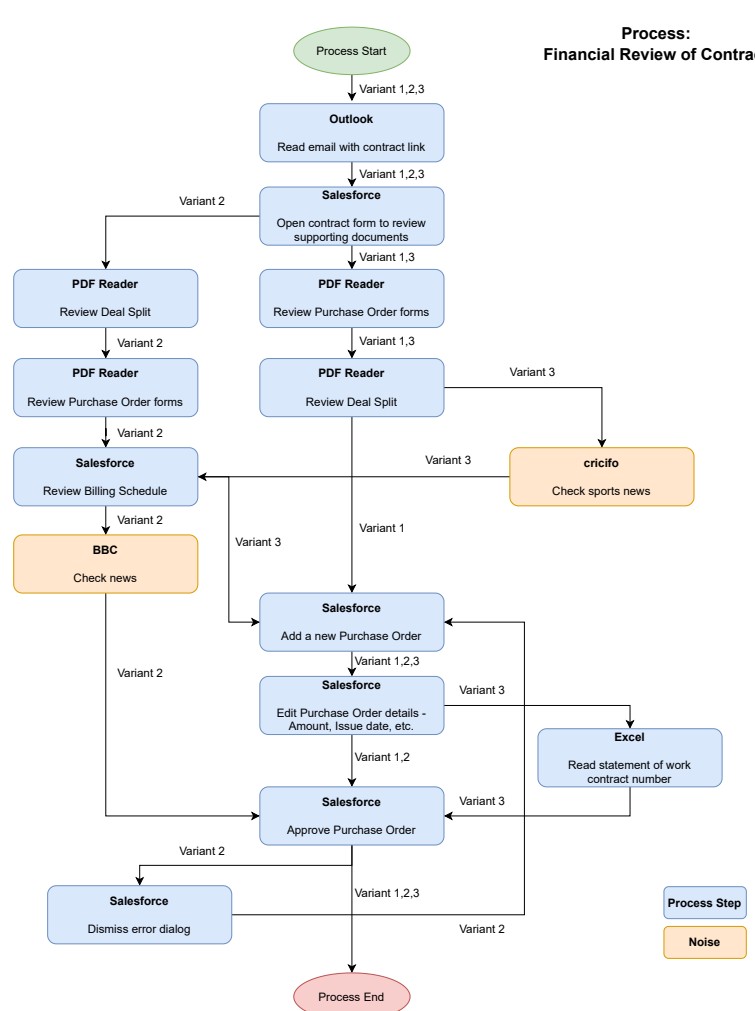

Figure 3: Three variants of the 'Financial review of contract' process. All flows involve Microsoft Outlook, Salesforce, and a PDF reader. Variants 1 and 3 include the creation of a purchase order, while Variant 2 only requires reviewing and approving an existing one. Variant 2 includes an error path due to a premature approval attempt. Users executing Variants 2 and 3 briefly visited unrelated websites (general news and sports), introducing natural noise into the recording.

details before resubmitting. During this process, the user briefly visits a news website, possibly due to software lag, introducing digital noise.

**Variant 3:** This variant begins similarly to the previous ones, with an email linking to the CRM system. The user first reviews the purchase orders, followed by the deal-split, and, as in Variant 2, also examines the billing schedule. A new purchase order is then created in the CRM based on the seller's terms, and the required data is entered. Before approving, the user consults a Statement of Work (SOW) in an Excel file, then finalizes the purchase order. During this process, the user briefly visits a sports website, again introducing natural noise, likely due to CRM lag.

The three process instances described above appear in the dataset with the following `process_instance_uuid`'s:

**Variant 1:** `4b15f7fc-22de-4c2e-bcd1-bb26888c9849`

**Variant 2:** `cfc96774-a5eb-4412-a7d5-ee1c66124d42`

**Variant 3:** `d3985ee7-d3f6-4143-9b9a-a726bc7b0c6c`

# B    BENCHMARK TASKS

## B.1    COMPUTE RESOURCES USED

All experiments were conducted using a combination of local GPU resources and cloud-based API endpoints, depending on the model provider. For local inference with open-source models (e.g., Qwen3-32B, DeepSeek-R1), we used an NVIDIA HGX H200 cluster (8 GPUs, 141GB VRAM each), running CUDA 12.8 and PyTorch 2.0. All experiments were orchestrated on a Linux server (Ubuntu 22.04, kernel 5.15), with 2TB RAM and 192 CPU cores, ensuring that data preprocessing and result aggregation did not bottleneck the evaluation pipeline. The batch size for local models was set to 8, as specified in the configuration files. GPU allocation can be managed from the configuration files (see `configs/classification.yaml` and `configs/segmentation.yaml` in our repository). For API-based models (OpenAI GPT-4.1, Anthropic Claude-3.5-Haiku, Google Gemini-2.5-Flash), requests were made via the respective Python SDKs, with a wait time of 1-15 seconds between requests to avoid rate limiting, as configured.

For embedding-based analyses (e.g., workflow similarity), we used the OpenAI `text-embedding-3-large` model via API, with a maximum context window of 8,000 tokens per request. To ensure reproducibility, all random seeds were fixed (seed=2404), and detailed logs of model configuration and evaluation metrics were maintained.

## B.2    ADDITIONAL EXPERIMENT DETAILS

For all model evaluations, we used a temperature of 0.6 to reduce the variance of the model's output. The maximum number of tokens to generate (`max_tokens`) was set to 5000 for classification tasks and 8000 for segmentation tasks, as specified in the respective configuration files (`configs/classification.yaml`, `configs/segmentation.yaml`).

**Prompt Templates:** The prompts provided to the models were carefully structured to elicit the desired outputs for each task. Placeholders in the templates were dynamically filled with relevant data for each evaluation instance.

**Classification Prompt:** The prompt for the workflow classification task was designed to be concise and direct, providing the model with the sequence of user interactions and the set of possible workflow classes. The structure, shown below, was:

```
Classification System Prompt

You are a workflow classification assistant that analyzes
user interactions and determines the workflow type.
```

```
Classification Prompt Template

Given the following user interaction sequence, classify it
into one of the following workflow types: {classes}.

User interaction sequence:
{sequence}

Provide your answer enclosed in \answer{}.
```

Here, `{classes}` was replaced with the actual class names for the domain, and `{sequence}` was replaced with the concatenated descriptions of all user interactions within the workflow instance.

**Segmentation Prompt:** For the workflow segmentation task, the prompt was more detailed, providing context about the task, definitions of possible workflow processes, the concatenated interaction sequence, and instructions for the JSON output format. The template, shown below, was:

**Segmentation System Prompt**

```
You are a workflow segmentation assistant that analyzes
sequences of user interactions and identifies where
different workflows begin and end.
```

**Segmentation Prompt Template**

```
Your task is to precisely segment a sequence of user
interactions that come from MULTIPLE WORKFLOWS
concatenated together.

Here are the workflow process definitions you should
consider: {process_definitions}

Your task:
1. Analyze the entire sequence carefully
2. Identify where one workflow ends and another begins
3. Mark the exact positions (indices) of these boundaries

User interaction sequence (0-indexed):
{sequence}

Step-by-step approach:
1. First, review the entire sequence to understand
   the overall pattern
2. Look for clear transitions between different workflows
3. Pay attention to workflow beginning and completion signals
4. When identifying a boundary, note its exact index position
5. Ensure all segments together cover the complete sequence

Format requirements:
- Provide a JSON array where each workflow segment has:
  - "start_index": starting position (0-indexed)
  - "end_index": ending position (inclusive, 0-indexed)
- First segment should always have start_index = 0
- Each segment's end_index should be exactly
  one less than the next segment's start_index
- Last segment's end_index should be the last index
  in the sequence
- The answer must be enclosed in <answer> tags

Example:
For a sequence with 3 workflows, a valid response might be:
<answer>
[
  {{"start_index": 0, "end_index": 5}},
  {{"start_index": 6, "end_index": 12}},
  {{"start_index": 13, "end_index": 17}}
]
</answer>

Before finalizing your answer:
- Verify that your segments correctly capture
workflow transitions
- Check that all indices are within the sequence bounds
- Confirm that segments are contiguous (no gaps or overlaps)
```

In this template, {process_definitions} was populated with the descriptions of all potential workflow processes for the given domain, and {sequence} contained the full sequence of user interactions to be segmented.

**Reproducibility and Data Availability:**  All code, configuration files, and scripts required to reproduce the experiments and generate the figures are available in our project repository included in the supplementary materials. The dataset, including all interaction sequences and workflow definitions, is provided in JSON format at https://www.kaggle.com/datasets/ 94647fd0bb51dff501a463674a2314627cdaf8c76d41b093c333b608459e017e. All plots in this appendix were generated using the provided scripts in src/generate_plots.py and are available in the figures/ directory.

### B.2.1 CLASSIFICATION TASK

For workflow classification, each model was evaluated in a strict zero-shot setting: the prompt included only the interaction sequence and the list of possible workflow classes, with no domain-specific examples or fine-tuning. The evaluation was performed across three domains: HR, Finance, and Legal. For each domain, we processed the entire dataset, feeding one workflow instance at a time to the model and recording the predicted class. The primary metric was accuracy, with additional reporting of per-class precision, recall, and F1-score.

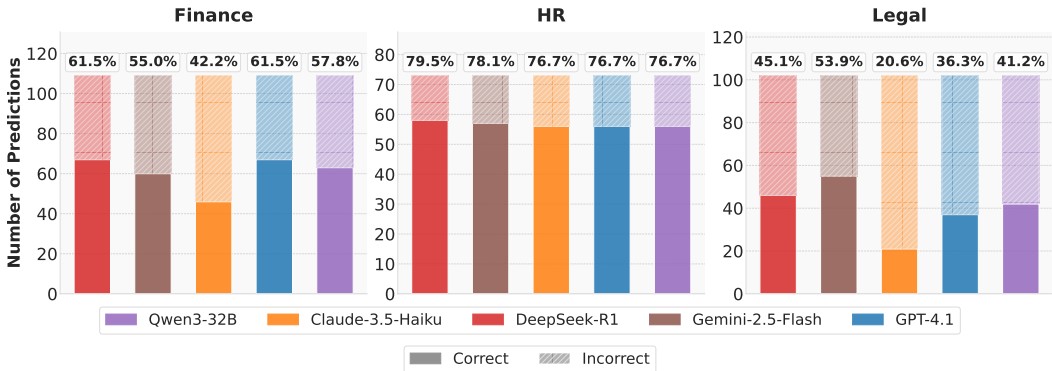

Figure 4: Per-domain classification accuracy of all evaluated models. Bar height indicates the number of workflow instances; bar fill (%) indicates correctness.

**Per-Class Accuracy:**   To provide a more granular view of performance within each domain, Figs. 5, 6, and 7 show the class-level accuracy for all evaluated models across the HR, Finance, and Legal domains, respectively. These plots highlight which specific workflow classes are well-recognized and which pose challenges for each model.

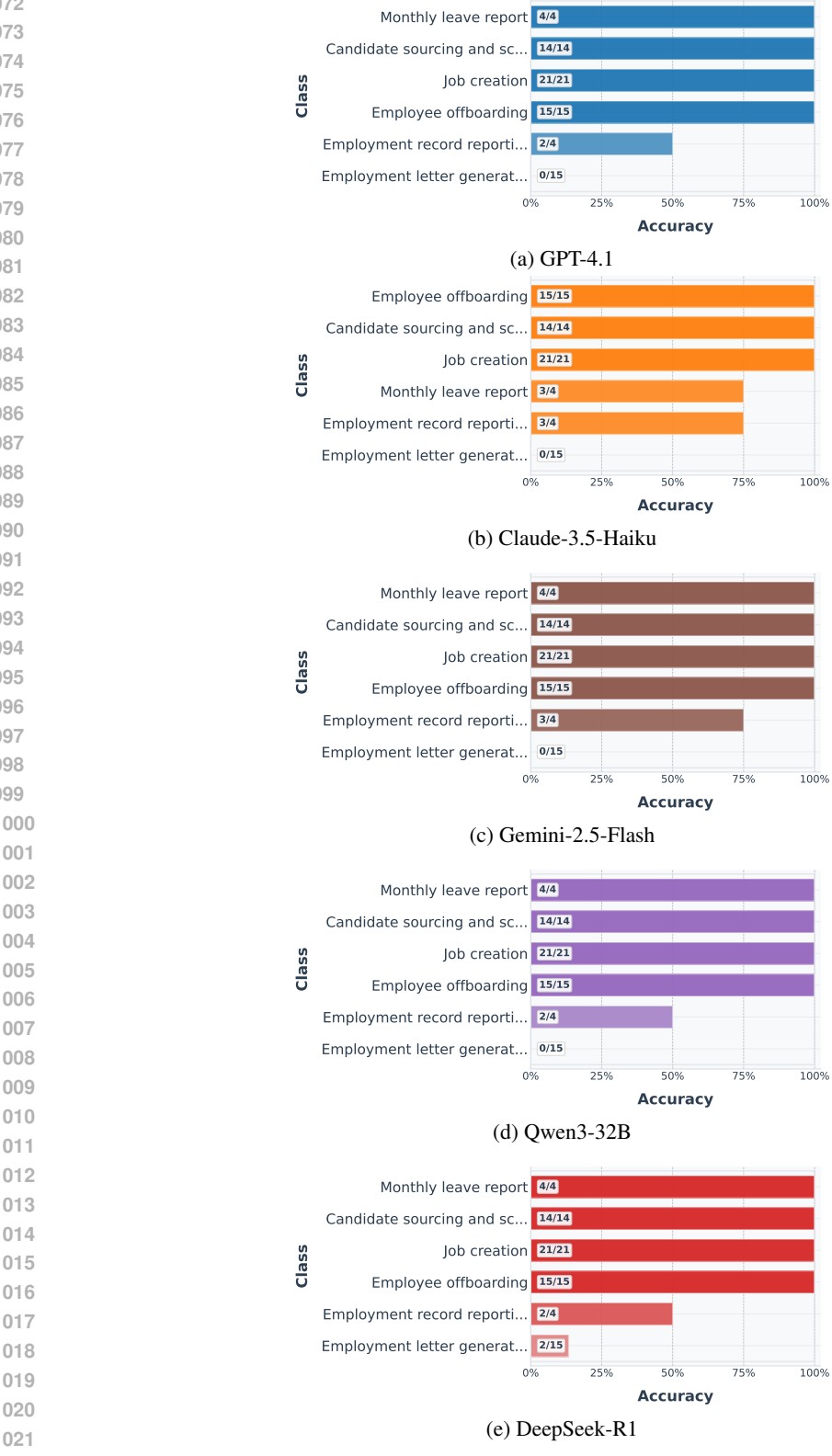

(a) GPT-4.1

(b) Claude-3.5-Haiku

(c) Gemini-2.5-Flash

(d) Qwen3-32B

(e) DeepSeek-R1

Figure 5: Class-level accuracy on HR workflows for all evaluated models.

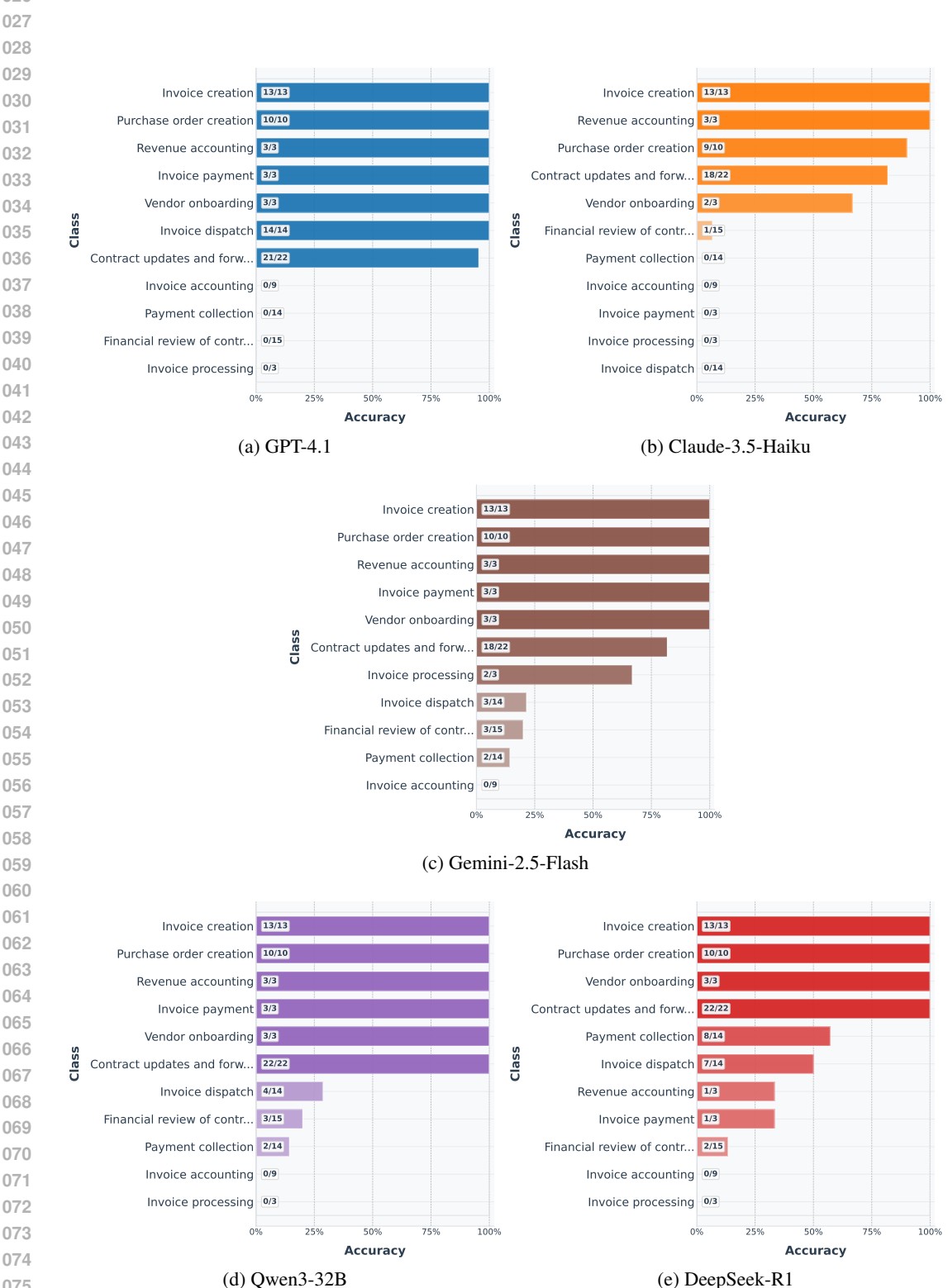

Figure 6: Class-level accuracy on Finance workflows for all evaluated models.

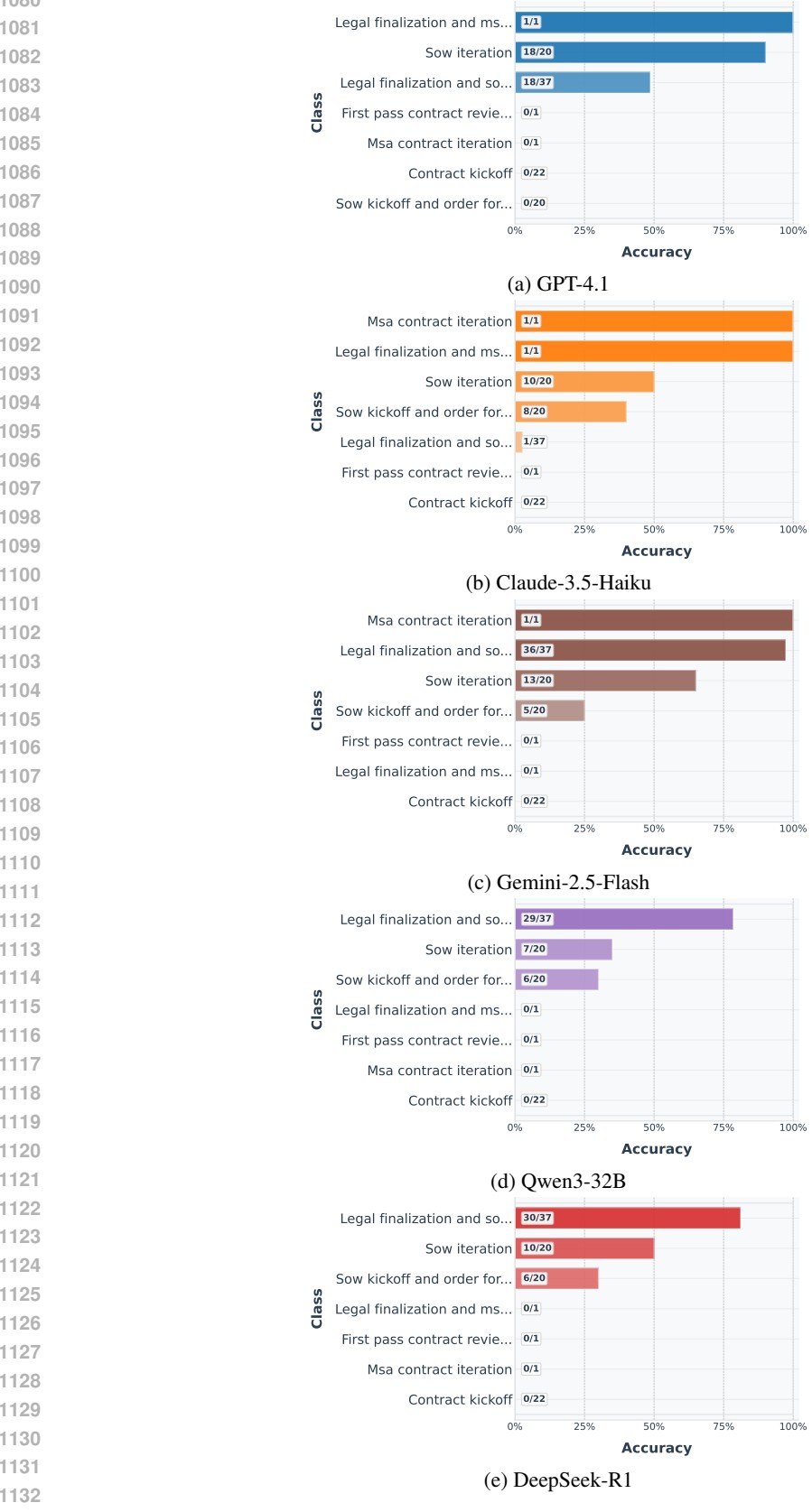

(a) GPT-4.1

(b) Claude-3.5-Haiku

(c) Gemini-2.5-Flash

(d) Qwen3-32B

(e) DeepSeek-R1

Figure 7: Class-level accuracy on Legal workflows for all evaluated models.

**Confusion Matrices:** Confusion matrices further detail the classification errors, illustrating the extent to which models confuse different workflow classes. Figs. 8, 9, and 10 present these matrices for all evaluated models across the HR, Finance, and Legal domains, respectively. These visualizations are particularly insightful for understanding misclassifications. For instance, the high degree of semantic overlap between certain Legal workflows, leading to model confusion, is evident across multiple models.

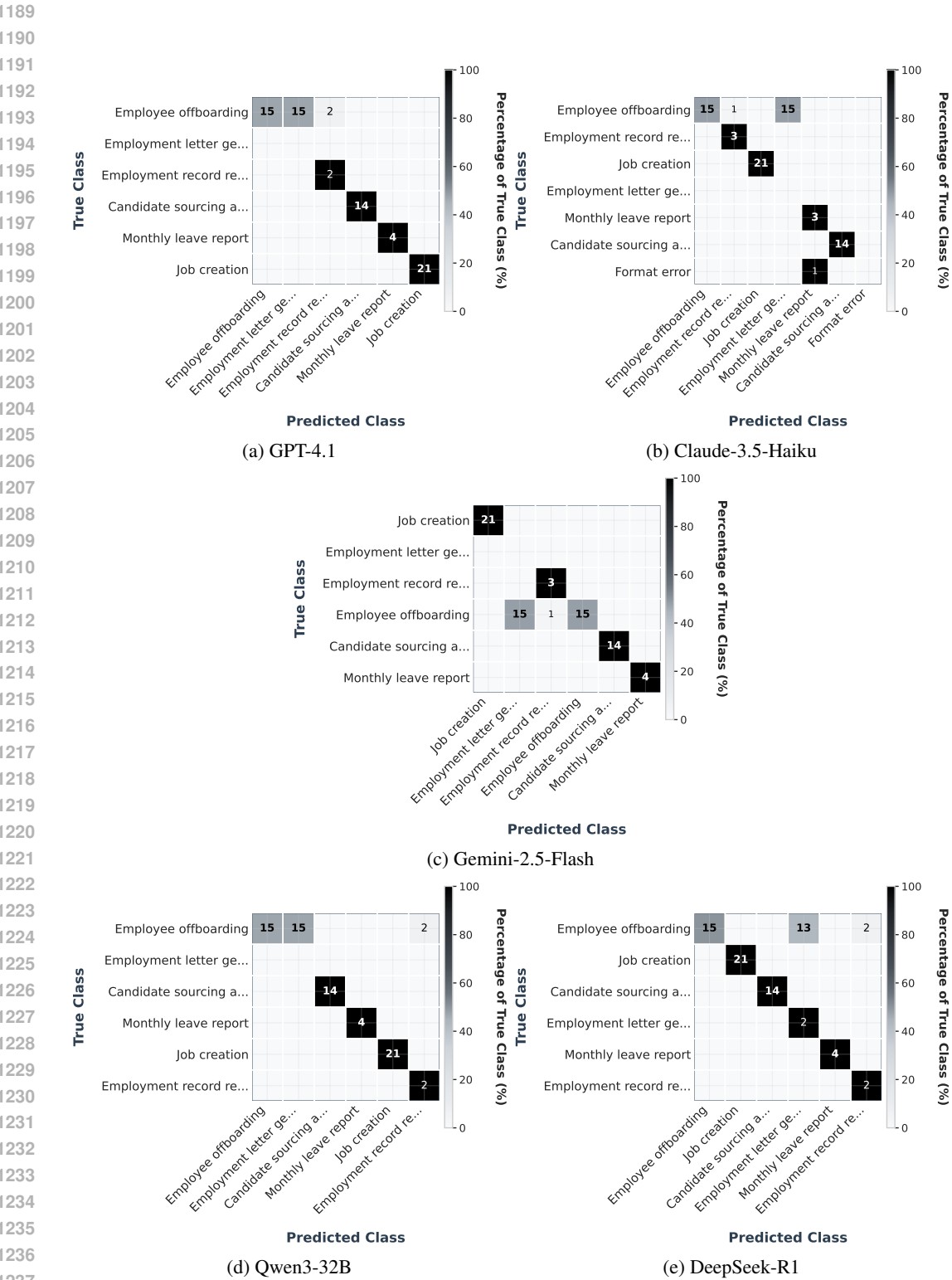

Figure 8: Confusion matrices on HR workflows for all evaluated models.

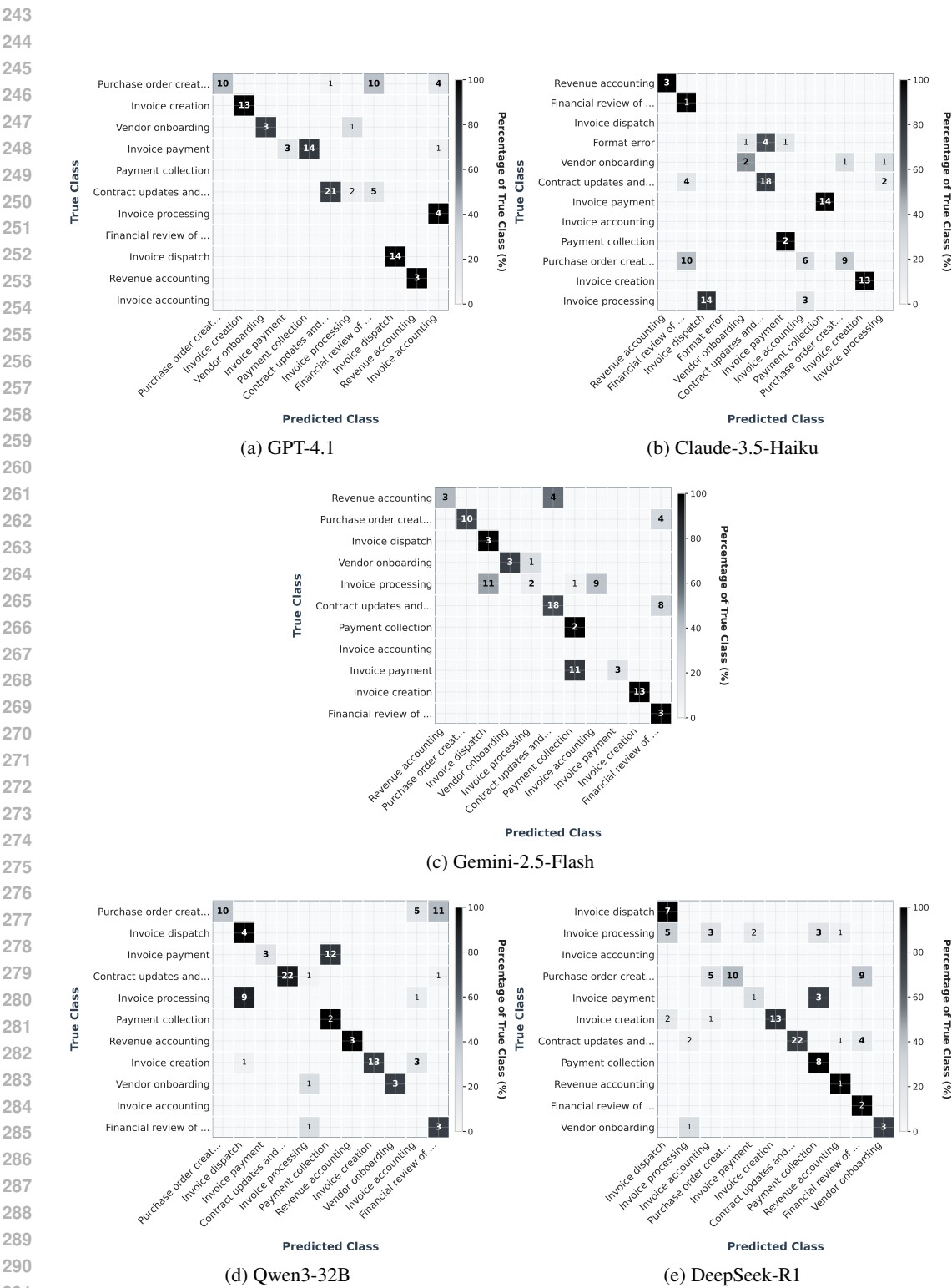

Figure 9: Confusion matrices on Finance workflows for all evaluated models.

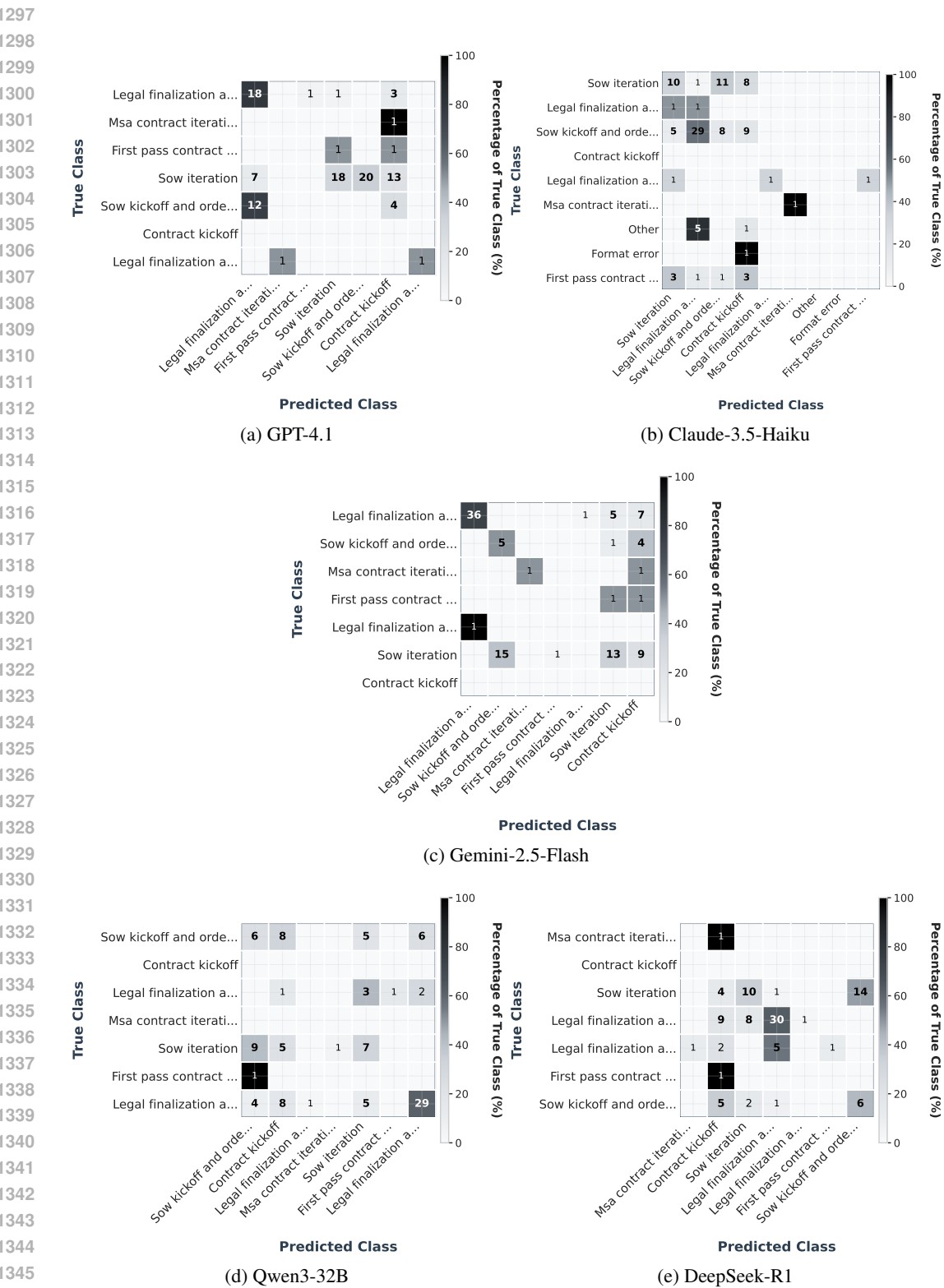

(a) GPT-4.1

(b) Claude-3.5-Haiku

(c) Gemini-2.5-Flash

(d) Qwen3-32B

(e) DeepSeek-R1

Figure 10: Confusion matrices on Legal workflows for all evaluated models.

**Aggregated Metrics:** Finally, overall classification metrics including precision, recall, and F1-score for each class are aggregated per domain. Figs. 11, 12, and 13 illustrate these aggregate metrics for the HR, Finance, and Legal domains, respectively. These plots provide a consolidated summary of the types of performance metrics collected across all classes within each domain. Detailed model-specific numerical results for these metrics are available in the JSON files provided in the supplementary materials.

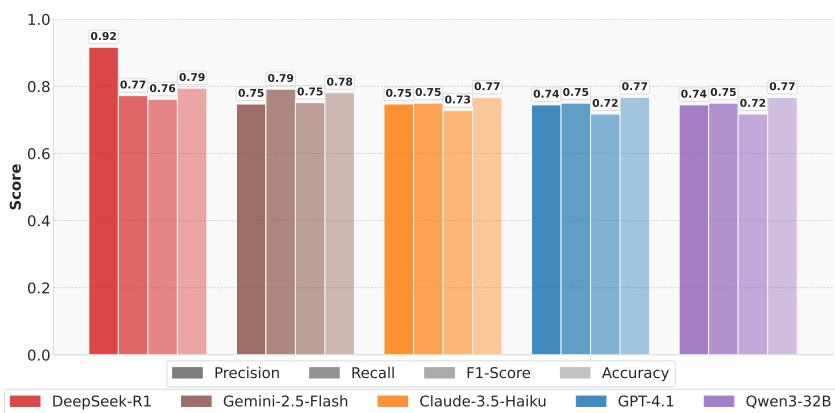

Figure 11: Aggregate classification metrics for HR workflows.

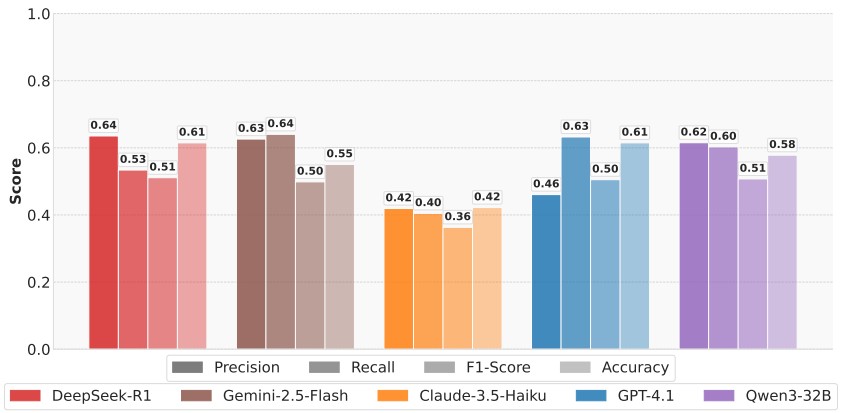

Figure 12: Aggregate classification metrics for Finance workflows.

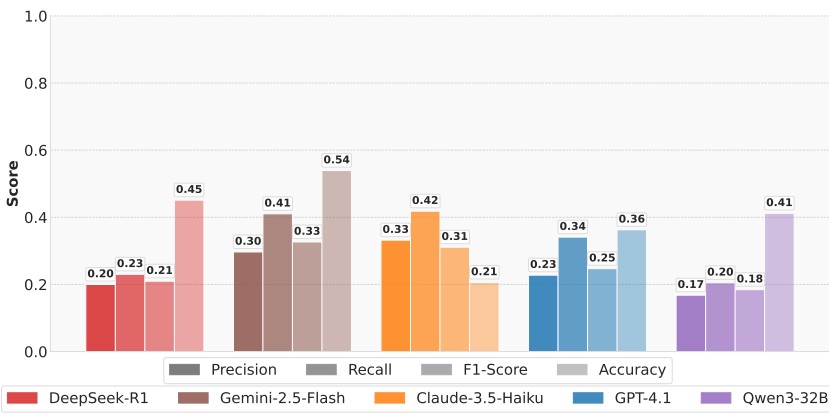

Figure 13: Aggregate classification metrics for Legal workflows.

### B.2.2 SEGMENTATION TASK

For workflow segmentation, we constructed 100 input samples per domain by concatenating 2-5 randomly selected workflow instances. Each model received the concatenated sequence of interactions and a list of all possible process definitions, and was prompted to output a JSON array of start/end indices for each segment. Evaluation metrics included boundary precision, recall, F1-score, and edit distance, computed by comparing predicted and true segment boundaries.

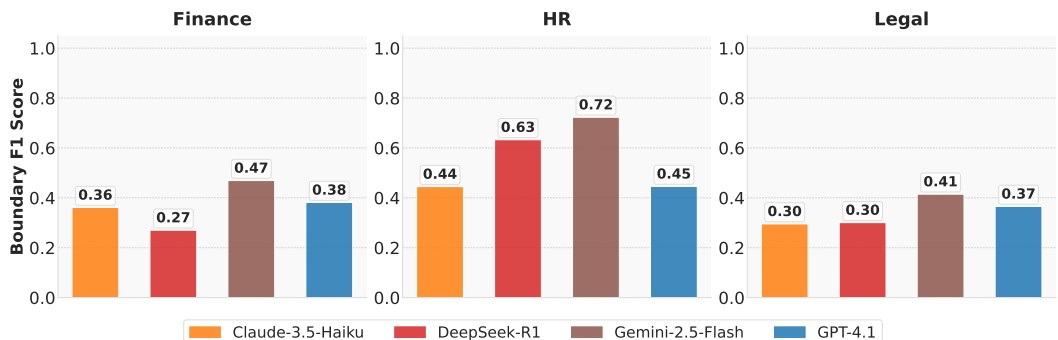

Figure 14: Zero-shot segmentation F1 scores for all models across domains.

The following figures provide a more detailed look at the boundary detection metrics for the segmentation task within each domain. Figs. 15, 16, and 17 display the distribution of F1 scores, precision, and recall for the HR, Finance, and Legal domains, respectively. These plots further illustrate the characteristics of model performance, such as the general tendency to over-segment (higher recall than precision), particularly in the more complex Finance and Legal domains.

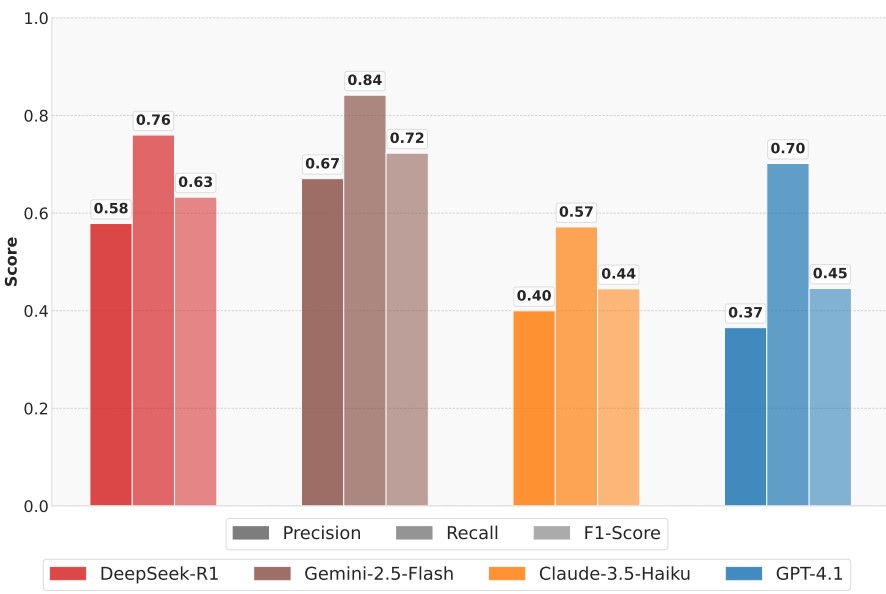

Figure 15: Distribution of boundary detection metrics for HR workflows.

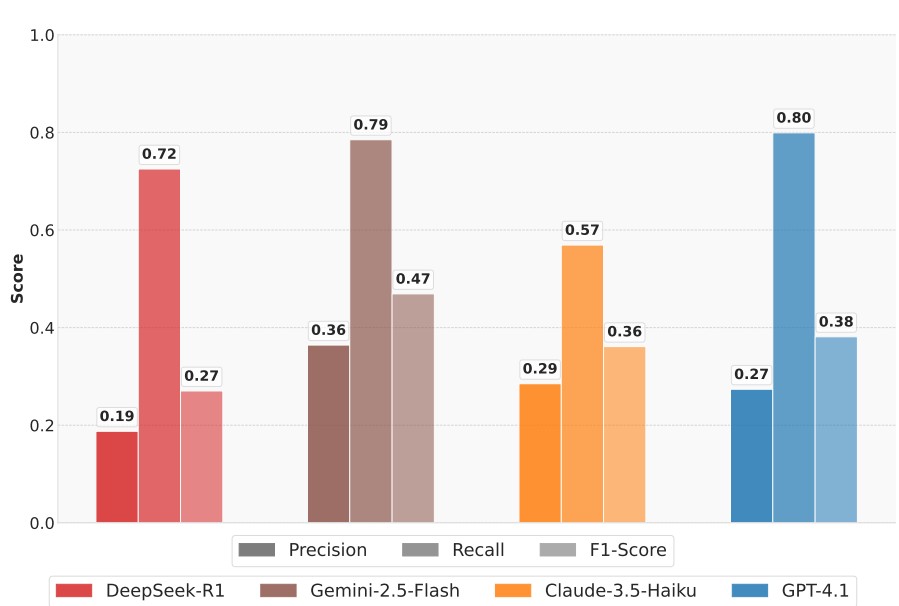

Figure 16: Distribution of boundary detection metrics for Finance workflows.

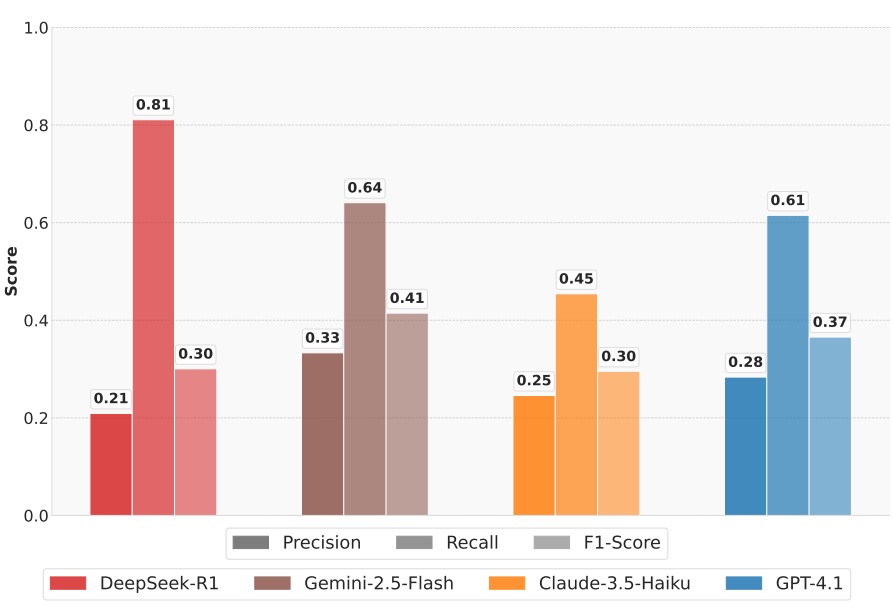

Figure 17: Distribution of boundary detection metrics for Legal workflows.

### B.3 MULTIMODAL EXPERIMENTS

Our primary benchmarks use only the structured digital interaction sequences from **ENTROPHY**. This choice ensures fair evaluation across a range of models, including widely used frontier models such as DeepSeek-R1 that do not support multimodal input. That said, **ENTROPHY** is inherently multimodal and pairs interactions with a screenshot of the user interface. To test whether visual context improves performance, we ran additional multimodal experiments with GPT-4.1.

Because workflows in **ENTROPHY** often span several hundred steps, including all screenshots would exceed provider token limits. To remain feasible, we downsampled the visual input by selecting a small set of representative frames. For workflow classification, we included 5 uniformly sampled screenshots per workflow. Accuracy improved from $61.5\% \rightarrow 67.6\%$ (finance), $76.7\% \rightarrow 78.1\%$ (HR), and $36.3\% \rightarrow 41.2\%$ (legal). For segmentation, we concatenated 2–3 workflows and added 3 screenshots per workflow. F1 scores improved from $0.38 \rightarrow 0.44$ (finance), $0.45 \rightarrow 0.46$ (HR), and $0.37 \rightarrow 0.38$ (legal). These modest but consistent gains suggest that screenshots provide complementary cues for both task recognition and boundary detection, though scalability remains limited by context window and token constraints.

We also evaluated UI-TARS (Qin et al., 2025), a UI-oriented model built on Qwen2.5-VL with a 32k-token context window. To fit within this window, we restricted inputs to the interaction sequences plus up to 5 screenshots per workflow. On the workflow classification task, UI-TARS achieved 40.7% (finance), 69.9% (HR), and 39.2% (legal). Segmentation was not feasible due to context length limitations. Overall, UI-TARS underperformed the frontier LLMs reported in the main paper, indicating that **ENTROPHY** poses a meaningful challenge even for current UI-specialized models.

In summary, multimodal inputs yield measurable accuracy gains, and UI-specific models still struggle with **ENTROPHY**'s long, variable workflows. These results highlight both the difficulty and the potential of **ENTROPHY** as a benchmark for developing more capable multimodal and workflow-oriented AI systems.

### B.4 PROCESS SIMILARITY ANALYSIS

We characterize the dataset by measuring the overlap between processes and their corresponding workflow executions, providing insight into the degree of similarity among processes within the same team. High pairwise similarity indicates that distinguishing between processes – whether for classification or segmentation – may be more challenging.

High similarity between processes or workflows often arises from shared use of the same applications, screens, fields, or documents. This results in overlapping digital interactions or only minor differences between them. For example, two processes that use the same application screen will appear similar, even if they interact with different fields. Without more advanced modeling, text-based embeddings will treat such processes as being closely related, making them harder to distinguish. For example, in legal workflows multiple phases may involve reviewing the same contract within the same applications, leading to similar digital traces.

To illustrate process similarity, we compute a pairwise similarity score between each process instance. Instances belonging to the same process are expected to exhibit a baseline level of similarity – e.g., greater than 0.7 – though not perfect (i.e., less than 1.0) due to natural variation in how the process is executed. However, high similarity between different processes and their instances, such as Legal's 'Contract kickoff' and 'First pass contract review', could indicate potential challenges for classification and segmentation, as previously discussed.

The similarity score between two process instances is computed as follows: Each process execution is converted into a paragraph, where each digital interaction is represented as a sentence. For example, 'The user clicked on the Submit button in the screen Invoice NetSuite of the application Oracle NetSuite'. This paragraph, representing the full sequence of digital interactions in the process execution, is then embedded using the *text-embedding-3-large* model, a state-of-the-art model for generating text embeddings. The resulting embedding serves as the representation of the process instance, and the pairwise similarity between any two instances is computed as the cosine similarity of their embeddings.

The pairwise similarity scores across all teams and their processes are shown in Fig. 18. Each point represents the cosine similarity between two process workflow executions. Darker boundaries are overlaid on the heatmap, with process-specific color labels along the border to indicate which executions belong to which process. This effectively groups the workflow executions by process and provides visual cues to help distinguish between them.

These results highlight several important characteristics that reflect the complexity and diversity of real-world enterprise workflows:

- **High similarity across distinct processes:** The Legal subplot illustrates a case where different processes exhibit high similarity (e.g., 0.75 or greater). For example, 'Contract kickoff' and 'First pass contract review' show substantial overlap. As discussed earlier, this is driven by factors such as the reuse of the same applications and documents across processes. Legal workflows often involve reviewing the same contract across multiple stages using common tools, leading to shared interaction patterns across distinct processes.

- **Low similarity within a process's executions – i.e., high variability in execution:** In contrast, HR processes reveal greater variability within individual workflows. For instance, the 'Employment letter generation' process shows lower within-process similarity (between 0.5 and 0.75), suggesting diverse execution paths or optional steps. Variability may stem from differences in employment types, roles, or regional requirements, all of which influence the specific workflow path followed in each instance.

- **High similarity within a single process – i.e., high standardization in execution:** Conversely, the 'Employee offboarding' process exhibits very high within-process similarity (0.9 or greater), indicating a highly standardized execution. This is expected for compliance-critical workflows like offboarding, where consistent application of policy and procedure is essential.

These results help explain the variance in model performance observed in the benchmark evaluations presented in the main body of the paper. Specifically, frontier models perform worse when classifying workflows within the Legal domain due to substantial overlap across its processes, e.g., due to reuse of the same or similar contracts, as illustrated in Fig. 18. In contrast, HR processes tend to be more distinct, allowing the same models – despite not being explicitly optimized for this dataset – to perform better in that domain. A similar trend is observed in the segmentation task, where reduced performance on Legal workflows can likewise be attributed to high structural similarity across process instances. These findings underscore the value of **ENTROPHY** in training models to disambiguate between processes with overlapping patterns.

Understanding these characteristics is important for analyzing process execution, which can vary significantly across teams and workflows. Metrics such as pairwise similarity across process instances offer a way to quantify this variation and provide insights that are critical for assessing the realism and quality of a dataset like **ENTROPHY**.

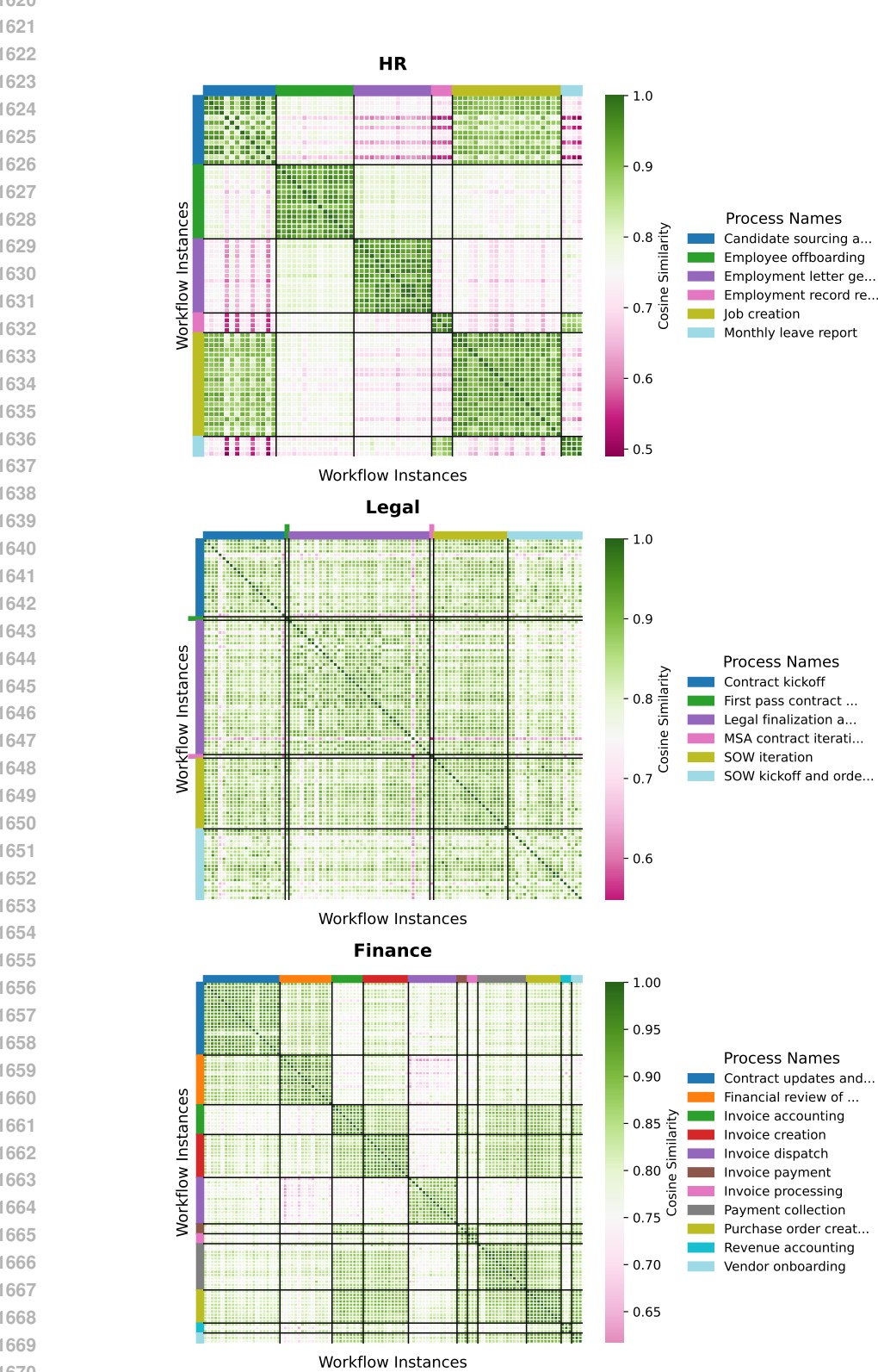

Figure 18: Pairwise similarity scores between process instances.

