# OpenReview forum: "Entrophy: User Interaction Data from Live Enterprise Workflows for Realistic Model Evaluation"
_ICLR.cc/2026/Conference — ICLR 2026 Conference Withdrawn Submission_

### Official Review · Reviewer_TgbV · 2025-10-31

**Soundness:** 3
**Presentation:** 2
**Contribution:** 2
**Rating:** 2
**Confidence:** 4

**Summary:**

The paper introduces ENTROPHY, a dataset of real-world enterprise workflows, capturing 33 hours of detailed digital interactions across finance, legal, and HR domains. Recorded from professionals performing authentic tasks, the dataset logs 283 workflow instances across 19 applications, integrating clicks, keystrokes, and screenshots. Benchmarking top LLMs on workflow classification and segmentation tasks shows limited accuracy, highlighting major challenges for AI automation in complex, real-world enterprise environments.

**Strengths:**

1. The benchmark is expertly labeled, with a clearly documented and rigorous annotation process.
2. The inclusion of realistic “noise” in the dataset enhances ecological validity and better reflects real-world enterprise workflows.

**Weaknesses:**

1. The experimental focus on classification and segmentation appears dated in the LLM era, where workflow generation and execution are more relevant research directions.
2. Given the dataset’s multimodal nature, the evaluation method for text-only models requires clarification.
3. The benchmark may mislead readers due to its resemblance to “entropy” in information theory.
4. The empirical findings in Section 5 are somewhat expected and offer limited novelty for the community.

**Questions:**

See the weakness part.

---

### Official Review · Reviewer_pjZk · 2025-11-01

**Soundness:** 3
**Presentation:** 3
**Contribution:** 3
**Rating:** 4
**Confidence:** 4

**Summary:**

This paper introduces a new dataset ENTROPHY  that contains real-world enterprise workflows. It was collected through fine-grained digital interaction logging (clicks, keystrokes, hotkeys) across 283 workflow instances, covering 33 hours of activity over 19 applications. The dataset spans finance, legal, and HR domains and captures structured workflow sequences. It benchmarks several LLMs on workflow classification and workflow segmentation tasks.

**Strengths:**

1. The paper targets an important problem in large language models, the workflow automation. Understanding and modeling enterprise workflows is a critical direction for building capable and trustworthy AI agents, and the authors make a notable effort to address this gap.
2. The dataset captures real-world, complex enterprise tasks that are rarely accessible to the research community. Collecting such data in authentic business environments is extremely challenging due to privacy, security, and compliance constraints. Thus, releasing ENTROPHY as an open dataset is a valuable contribution that will likely stimulate further research in realistic workflow modeling.
3. The paper provides a clear and comprehensive description of the dataset construction pipeline, data composition, and domain distribution.

**Weaknesses:**

1. While the dataset is positioned as a benchmark for workflow-related research, the paper only explores two downstream tasks, workflow classification and workflow segmentation. From an LLM and agentic research perspective, the community is increasingly interested in whether models can autonomously construct or generate workflows from natural instructions or demonstrations (e.g., [1, 2]). Evaluating LLMs solely on recognition or segmentation tasks underutilizes the dataset’s potential.
2. Closely related to the previous point, the paper does not introduce or provide an executable environment or workflow execution interface that could support generative workflow construction or end-to-end task completion. Without such a platform, the dataset’s applicability to studying workflow synthesis, planning, or tool orchestration remains limited.

[1] WorkflowLLM: Enhancing Workflow Orchestration Capability of Large Language Models. ICLR 2025.

[2] Generalizing Experience for Language Agents with Hierarchical MetaFlows. NeurIPS 2025.

**Questions:**

Suggestions for Improvement:

1. Consider adding a workflow construction or synthesis benchmark, where models must generate executable workflow representations or action sequences given textual task descriptions or partial demonstrations. Such an extension would make ENTROPHY more relevant for the current LLM-agent community and better connect with ongoing work in tool-using agents and process automation.
2. Optionally, the authors could also describe plans for an evaluation environment or simulation layer that allows future research on workflow execution or reinforcement learning over this dataset.

---

### Official Review · Reviewer_nWZH · 2025-11-01

[review text omitted: it was posted to a different submission]

---

### Note · Authors · 2025-12-04

I have read and agree with the venue's withdrawal policy on behalf of myself and my co-authors.